# Structure of HIV-1 RRE stem-loop II identifies two conformational states of the high-affinity Rev binding site

Jerricho Tipo [1,2], Keerthi Gottipati [2], Michael Slaton[2], Giovanni Gonzalez-Gutierrez [2] & Kyung H. Choi [1,2,3] ✉

During HIV infection, specific RNA-protein interaction between the Rev response element (RRE) and viral Rev protein is required for nuclear export of intron-containing viral mRNA transcripts. Rev initially binds the high-affinity site in stem-loop II, which promotes oligomerization of additional Rev proteins on RRE. Here, we present the crystal structure of RRE stem-loop II in distinct closed and open conformations. The high-affinity Rev-binding site is located within the three-way junction rather than the predicted stem IIB. The closed and open conformers differ in their non-canonical interactions within the three-way junction, and only the open conformation has the widened major groove conducive to initial Rev interaction. Rev binding assays show that RRE stem-loop II has high- and low-affinity binding sites, each of which binds a Rev dimer. We propose a binding model, wherein Rev-binding sites on RRE are sequentially created through structural rearrangements induced by Rev-RRE interactions.

During HIV-1 infection, the 9 kb positive-sense RNA genome is reverse-transcribed and integrated into the host cell's genome as proviral DNA[1–3]. Transcription of the provirus DNA results in the generation of fully spliced, partially spliced, and unspliced mRNA transcripts[4]. At the early stage of HIV infection, only the intron-free fully spliced transcripts exit the nucleus for translation into HIV regulatory proteins, Tat, Rev, and Nef[5–8]. Tat (trans-activator of transcription) recognizes its RNA-binding partner, the trans-activation response element (TAR), and enhances transcription of the full-length HIV RNA[9–11]. However, the partially spliced and unspliced transcripts contain intronic sequences and are thus retained in the nucleus. Rev (regulator of viral expression) then specifically recognizes the Rev response element (RRE), present in the partially spliced and unspliced HIV transcripts, and facilitates nuclear export of these incompletely spliced viral transcripts[12–15].

The ~350 nucleotide (nt) long RRE is highly structured and predicted to fold into four or five stem-loops (SL), SLI–V[16–19]. Two Rev-binding sites have been identified within RRE, a high-affinity site within stem-loop II (SLII) and a secondary binding site within stem I. SLII forms a three-stem structure, IIA-C, and a purine-rich internal loop within stem IIB is predicted to be the initial high-affinity Rev-binding site (Fig. 1A)[20,21]. The second Rev-binding site in stem I is also a purine-rich internal loop (designated IA) and is suggested to stabilize Rev multimers[22,23]. Stem-loops III–V make relatively minor contributions to RRE activity, and even a construct that lacks all three stems (III/IV/V) retains ~33% of the wild-type (WT) RRE activity[24]. Rev initially binds to the internal loop within stem IIB, which then triggers the oligomerization of as many as eight Rev molecules onto RRE[20,25–27]. Rev oligomerization is required for the biological activity of the Rev−RRE complex, as binding of a single Rev molecule is insufficient for nuclear export[28,29]. Upon formation of the homo-oligomeric Rev−RRE ribonucleoprotein (RNP), the C-terminal nuclear export signal of Rev recruits cellular Chromosomal Maintenance 1 (CRM1, also known as Exportin 1), and RAS-related nuclear protein to form the nuclear export complex[30–35]. This nuclear export complex shuttles the intron-containing viral transcripts to the cytoplasm, where the viral RNA is

[1]Department of Pharmacology and Toxicology, The University of Texas Medical Branch, Galveston, TX 77555, USA. [2]Department of Molecular and Cellular Biochemistry, Indiana University, Bloomington, IN 47405, USA. [3]Department of Biochemistry and Molecular Biology, Sealy Center for Structural Biology, The University of Texas Medical Branch, Galveston, TX 77555, USA. ✉e-mail: kaychoi@iu.edu

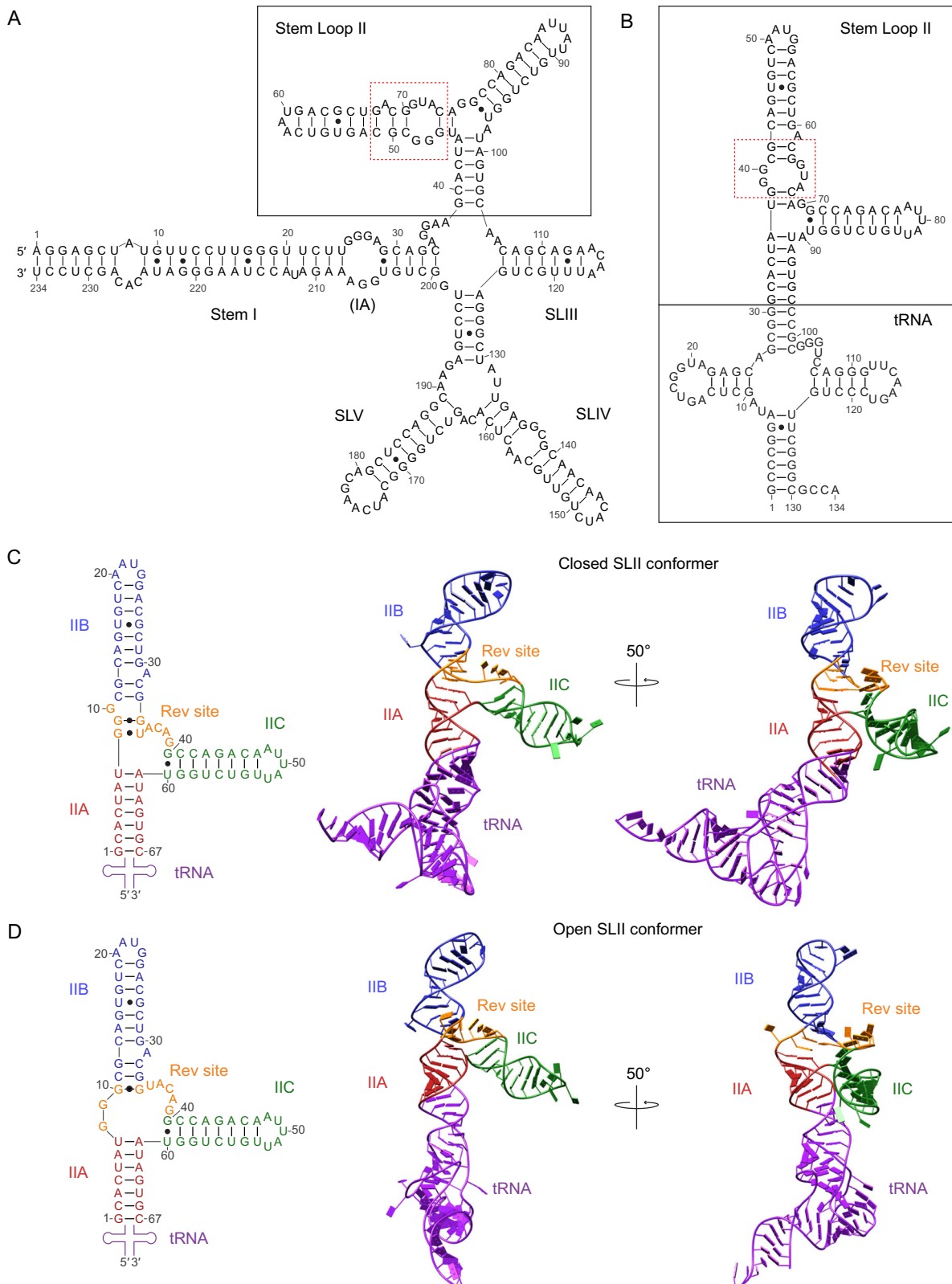

**Fig. 1 | Structures of the Rev response element (RRE). A** Secondary structure of the predicted four-stem RRE RNA. Stem-loop II (SLII) is boxed in black, and the high-affinity Rev-binding site is indicated with red dashed lines. **B** Design of SLII–tRNA. The SLII sequence is inserted into the anticodon stem of tRNA. Stem IIA is fused to the tRNA scaffold. The high-affinity Rev-binding site is indicated with red dashed lines. An additional guanine nucleotide (G53) was incorporated into stem IIB to engineer a tetraloop. **C** Structure of the SLII–tRNA in the closed conformation. The secondary structure (left) and the crystal structure (right) are shown. SLII consists of stems IIA (red), IIB (blue), and IIC (green). The high-affinity Rev-binding site (the triple guanine motif and junction nucleotides between IIB and IIC) is shown in orange. The tRNA scaffold is shown in purple. **D** Structure of the SLII–tRNA in the open conformation. The secondary structure (left) and the crystal structure (right) are shown. The SLII stems and the high-affinity Rev-binding site are colored as in (**C**).

**Table 1 | Anisotropic data collection and structure determination**

| | SLII–tRNA |
|---|---|
| **Data collection** | |
| Wavelength (Å) | 1.00003 |
| Space group | P2₁ |
| Molecules/AU | 2 |
| Cell dimensions | |
| a, b, c (Å) | 67.8, 81.3, 81.4 |
| α, β, γ (°) | 90, 99.1, 90 |
| Resolution (Å) | 57.2–2.85 (3.20–2.85)[a] |
| No. of unique reflections | 12,674 (635) |
| $I/\sigma I$ | 8.9 (1.0) |
| $R_{meas}/R_{p.i.m}$ | 0.10/0.05 (1.36/0.72) |
| $CC_{1/2}$ | 0.999 (0.487) |
| Completeness spherical (%) | 61.5 (11.8) |
| Completeness ellipsoidal (%) | 88.5 (68.3) |
| Redundancy | 4.0 (3.5) |
| **Refinement** | |
| Resolution (Å) | 57.2–2.85 (3.13–2.85) |
| $R_{work}/R_{free}$ | 0.22/0.25 (0.33/0.37) |
| No. of atoms | |
| Nucleotide | 5743 |
| R.M.S. deviations | |
| Bond length (Å) | 0.003 |
| Bond angles (°) | 0.735 |
| Mean B-factors (Å²) | |
| Nucleotides | 121.13 |

Single crystal was used for data collection.

[a]Values in parentheses correspond to the highest resolution shell.

used for the translation of late-stage viral proteins and assembly of viral particles[9–11].

Nuclear export of HIV viral transcripts is essential for progression to later stages of the HIV life cycle, yet the mechanism by which RRE is recognized by Rev is not well understood. In particular, it is not clear how Rev discriminates between RRE-containing RNA and other RNAs, or how multiple copies of Rev assemble on RRE given that only two binding sites exist on RRE. Structural studies of the Rev–RRE complexes have been limited to minimized stem IIB hairpin RNAs with Arginine-rich motif (ARM) peptide or N-terminal domain of Rev[7,20], which do not fully recapitulate the three-way junction structure of SLII. It is also not clear how many Rev molecules bind SLII following the initial Rev binding within stem IIB. Gel shift assays show that the full-length SLII can bind at least three Rev molecules[36,37], whereas native mass spectrometry shows that SLII interacts with up to five Rev ARM peptides[38].

To better understand how the full-length SLII structure affects Rev recognition and oligomerization, we determined the X-ray crystal structure of RRE SLII and performed binding studies with SLII RNA mutants. We observe that the native SLII structure exists in distinct "closed" and "open" conformations that differ in the non-canonical base pairs in the high-affinity Rev-binding site. We show that initial Rev dimer binding occurs at the high-affinity binding site within the three-way junction, and subsequent Rev dimer formation occurs in the adjoining stem structures of SLII. Based on our results, we propose a sequential binding model of Rev–RRE, in which the conformational state of RRE SLII acts as a checkpoint to direct Rev binding and oligomerization.

## Results

### RRE SLII forms a lambda (λ)-shaped three-way junction structure and the high-affinity Rev-binding site is located in the three-way junction

The crystal structure of RRE SLII was determined using the tRNA-scaffold approach[39,40], where the 67 nt of HIV-1 RRE SLII replaced the anticodon loop of the human tRNA[Lys] (134 nt total). The open stem IIA was fused to the anticodon arm of tRNA, maintaining 5′ and 3′ directionality (Fig. 1A, B). To facilitate crystallization, a guanine nucleotide (G23) was added to the apical loop of stem IIB to form a tetraloop structure (5′-AAU<u>G</u>-3′). Mutations in the apical loop of stem IIB do not affect Rev function[41]. The RNA crystallized in space group P2₁ with two molecules in the asymmetric unit, and the structure was determined to 2.85 Å resolution (Fig. S1). Data collection and refinement statistics are shown in Table 1.

The two RRE SLII molecules in the crystallographic asymmetric unit adopted two different conformations, one "closed" and the other "open". Both molecules are "lambda (λ)"-shaped and consist of stems IIA, IIB, and IIC connected by a three-way junction (Fig. 1C, D). Stem IIB is coaxially stacked on stem IIA, and stem IIC projects out from the three-way junction at a ~70° angle to stem IIA. The structures belong to the family A three-way junctions, where helix 3 (i.e., stem IIC) is roughly perpendicular to the coaxially stacked helices 1 and 2 (stems IIA and IIB) and the longest junction is between helices 2 and 3[42]. The predicted secondary structure of RRE SLII (Fig. 1A) was not correct near the three-way junction. In particular, the predicted internal loop in stem IIB (Fig. 1A) does not form. First, U[7] is a part of stem IIA, where it base pairs with A[61], instead of the predicted U[7]–A[39] base pair in stem IIB. Thus, stem IIA is one base pair longer than predicted. Second, the predicted G[8]–C[37] base pair in stem IIB does not form. As a result, the [36]ACAG[39] sequence is not incorporated into an internal loop and is instead single-stranded in the three-way junction (Fig. 1C, D). Thus, the three-way junction is much longer than predicted. Lastly, the non-Watson–Crick base pairs within stem IIB differ in the closed and open conformations. The high-affinity Rev-binding sequence, the triple guanine ([8]GGG[10]) motif, is incorporated into dsRNA at the base of stem IIB in the closed form; in the open form, only G[10] is incorporated into stem IIB while [8]GG[9] is single-stranded in the three-way junction (Fig. 1C, D) (see next section for details). As a result, the three-way junction in the open conformation has three more nucleotides (G[8], G[9], U[35]) compared to the closed form (Fig. 1C, D). It should be noted that the secondary structures of RRE determined via SHAPE studies disagree at the high-affinity Rev-binding site in SLII, and range from a closed internal loop to an open single-stranded structure[17,19] (Fig. S2). The observed two conformations of SLII explain this discrepancy, as the closed SLII conformer is similar to the closed secondary structure[17] and the open SLII conformer resembles that of the open secondary structure[19] (Fig. S2).

### Non-canonical base interactions within the three-way junction of RRE SLII differ in the closed and open conformations

The high-affinity Rev-binding site in RRE has been identified by RNase protection assays in the presence of Rev[43]. Rev protects the [8]GGG[10] motif, [33]GG[34], and A[36] nucleotides within SLII (SLII numbering, Fig. 1), all of which are located in the three-way junction of the SLII crystal structures. The [8]GGG[10] motif in RRE SLII has also been shown to be essential for Rev binding and activation of Rev function in cells[43]. Deletion of one G from the GGG motif resulted in only ~40% of Rev response, and deletion of two or all three G's led to a complete loss of Rev binding and Rev response[43]. Similarly, the substitution of [33]GG[34] to CC resulted in a complete loss of Rev binding and Rev response in cells. The closed and open conformers differ in non-canonical base pair interactions within the initial high-affinity Rev-binding site ([8]GGGCG[12]/[32]CGGUACA[38], Figs. 1 and 2). In the closed conformation,

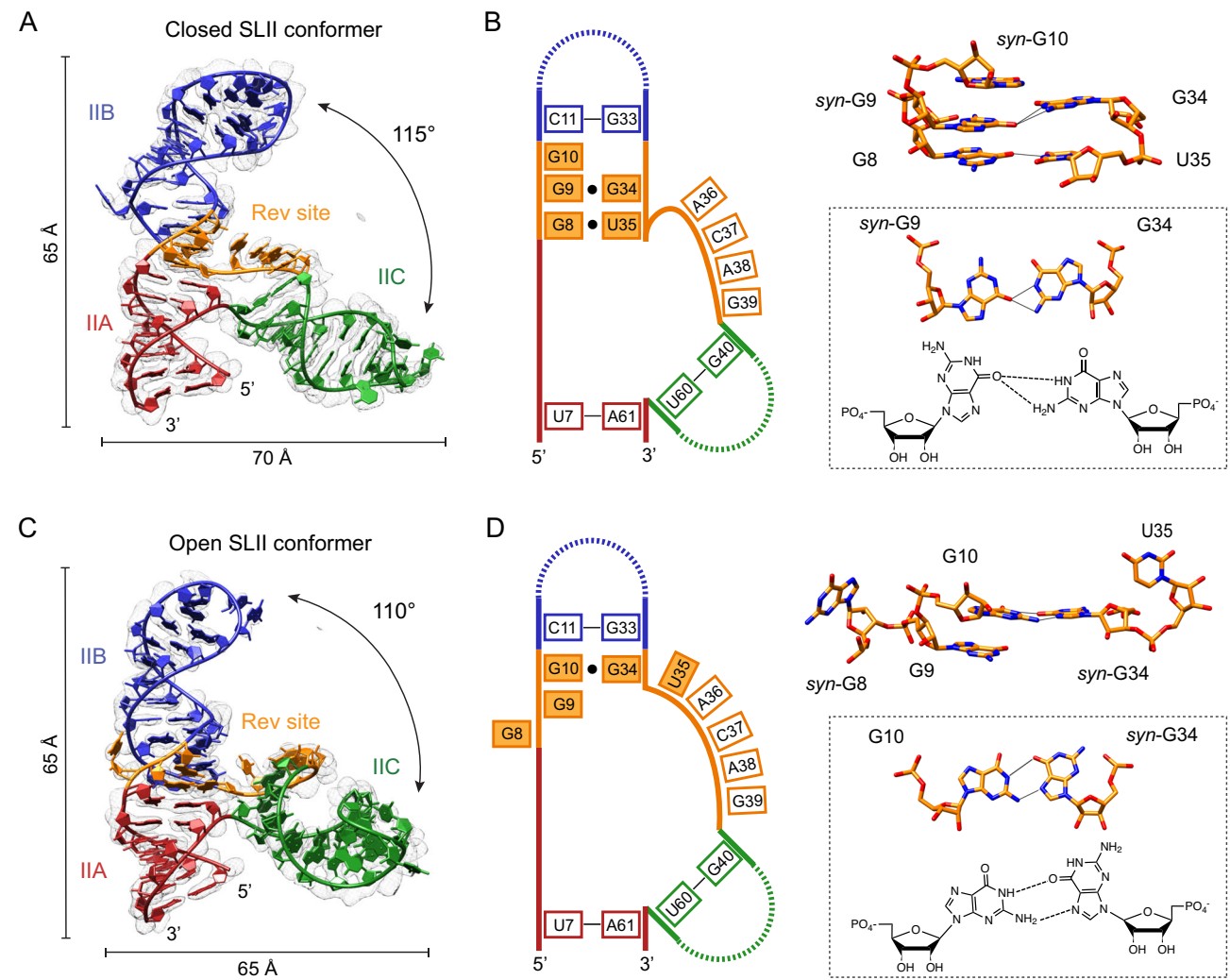

**Fig. 2 | Interactions within the triple guanine (GGG) motif of the closed and open SLII conformers. A** Closed RRE SLII conformer. The SLII closed conformer is shown with the $2F_oF_c$ electron density contoured at 1.0 σ in light gray. The size of the molecule and the angle between stems IIB and IIC are indicated. **B** Tertiary interactions in the high-affinity Rev-binding site of the closed conformer. A schematic of base pair interactions in the three-way junction are shown (left). The filled boxes indicate the nucleotides involved in the interactions mediating non-canonical base pairs and the opening/closing of the three-way junction. The close-up view of the high-affinity Rev-binding site (the GGG motif, filled boxes) is shown (right). Note that closure of the three-way junction, and thus lengthening of stem IIB, is mediated by non-canonical $G^8 \cdot U^{35}$ and $syn$-$G^9 \cdot G^{34}$ base pairs. The $syn$-$G^{10}$ base

stacks on the $syn$-$G^9 \cdot G^{34}$ base pair. The $syn$-$G^9 \cdot G^{34}$ base pair interaction is shown along with chemical structures. Hydrogen bond lengths below 3.4 Å are displayed. **C** Open RRE SLII conformer. The SLII open conformer is shown with the $2F_oF_c$ electron density contoured at 1.0 σ in light gray. **D** Tertiary interactions in the high-affinity Rev-binding site of the open conformer. Schematic of base pair interactions (left) and close-up views of the high-affinity Rev-binding site (the GGG motif, right) are shown. Note that only G34 forms a non-canonical $G^{10} \cdot syn$-$G^{34}$ base pair, thus opening the three-way junction. The $G^{10} \cdot syn$-$G^{34}$ base pair interaction is shown along with their chemical structures. The structures and schematics are colored as in Fig. 1.

the A-form helical structure is maintained at the triple guanine motif ($^8GGG^{10}$) via non-canonical base-pairing interactions of $G^8 \cdot U^{35}$ and $G^9 \cdot G^{34}$ (Fig. 2A). Interestingly, $G^9$ is in the higher-energy $syn$ conformation and forms a non-canonical purine–purine base pair with $G^{34}$, where the Hoogsteen edge of $syn$-$G^9$ and the Watson–Crick face of $G^{34}$ interact (Fig. 2A, B). The $syn$-$G \cdot anti$-$G$ pairing pattern has been observed in other RNA structures, as it minimizes the purine–purine clash in the center of the duplex, while utilizing its hydrogen bonding potential[44]. $G^{10}$ base is also in the $syn$ conformation ($syn$-$G^{10}$) and stacks on $syn$-$G^9$ (Fig. 2B). Rotation of the guanine base about the glycosidic bond into the $syn$ state is thus compensated by stabilizing the overall tertiary structure of the RNA by stacking and base pairing[45]. As a result, the three-way junction of the closed form consists of single-stranded

$^{36}ACAG^{39}$ between stems IIB and IIC. The $^{36}ACAG^{39}$ stretch is stacked on one another and is stabilized by interaction with the engineered tetraloop on stem IIB ($^{20}AAUG^{23}$) of a symmetry-related molecule. The $^{37}CA^{38}$ base pair with $^{22}UG^{23}$ of the IIB tetraloop, and $G^{39}$ (sugar edge) forms hydrogen bonds with $A^{21}$ of the IIB tetraloop (Fig. S3).

The open conformation is marked by a non-canonical purine–purine base pair between $G^{10}$ and $G^{34}$ and melting of the $G^8 \cdot U^{35}$ base pair interactions at the base of stem IIB (Fig. 2C, D). Thus, stem IIB in the open conformation is shorter than that in the closed form. The $syn$-$G^9$ and $syn$-$G^{10}$ seen in the closed conformation are flipped into the standard $anti$ conformation, whereas $anti$-$G^{34}$ is flipped to the $syn$ conformation. $G^{10}$ of the $^8GGG^{10}$ motif forms a non-Watson–Crick base pair with $syn$-$G^{34}$, where the Watson–Crick face of

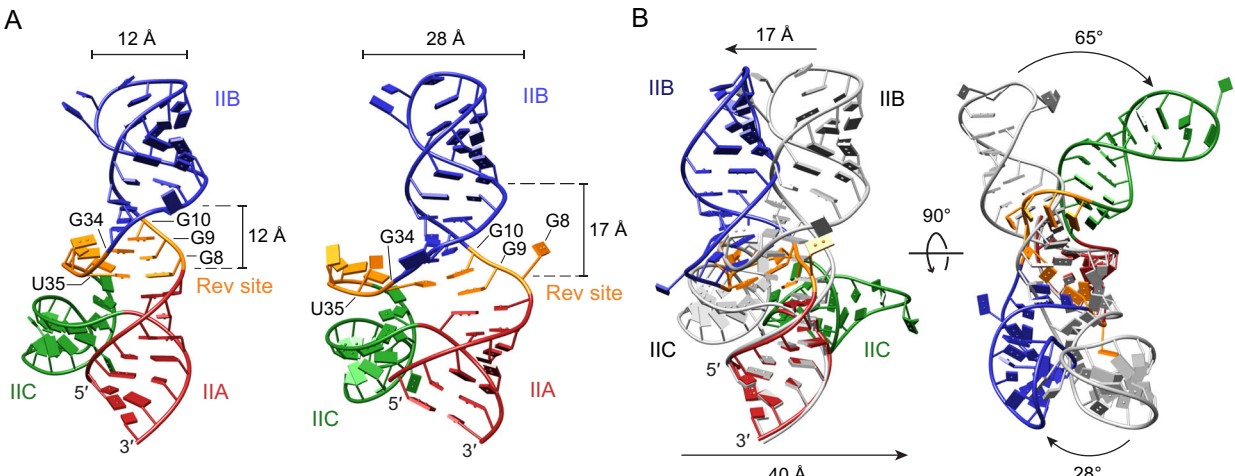

**Fig. 3 | Comparison between the closed and open SLII conformers.**
**A** Comparison of the RNA major grooves in the closed and open SLII conformers. In the closed conformer (left), the width of the major groove formed between stems IIA and IIB at the GGG motif is ~12 Å, and the distance between the C1' atoms of G8 and U35 is ~12 Å. In the open SLII conformer (right), the same major groove is widened to ~17 Å, and the distance between the C1' atoms of G8 and U35 is widened to ~28 Å. **B** Alignment of the two conformations at stem IIA. The closed conformer is colored in gray, and the open conformer is colored as in (**A**). The superposition of closed and open conformers with IIA indicates that opening of the high-affinity Rev-binding site rotates stems IIB by ~28° and IIC by ~65°.

$G^{10}$ forms hydrogen bonds with the Hoogsteen edge of *syn*-$G^{34}$ (Fig. 2D). The rest of the GGG motif ($^8GG^9$) and $^{35}UACAG^{39}$ are single-stranded in the three-way junction. $G^8$ and $U^{35}$ are flipped out of the helix and $G^9$ is stacked between $G^{10}$ and the $U^7$–$A^{61}$ base pair (stem IIA). The $^{37}CAG^{39}$ in the single-stranded region between stems IIB and IIC interact with $^{21}AUG^{23}$ in the stem IIB tetraloop of a neighboring molecule, similar to the closed conformation (Fig. S3).

The different non-canonical interactions in the closed and open forms alter the arrangement of the three stems, pivoting stems IIB and IIC at the three-way junction. In the closed conformer, the axes of the three stems are approximately in the same plane (Fig. 3A). Opening of the three-way junction rotates stems IIB and IIC out of this planar configuration. When stem IIA of the two conformers is superposed, stem IIB in the open form is related by ~28° to that of the closed form, resulting in ~17 Å translation at the tip of the apical loop (Fig. 3B). Stem IIC in the open conformer is related by 65° to that of the closed conformer, resulting in ~40 Å shift at the tip of its apical loop (Fig. 3B). The closed and open forms of full-length RRE SLII superpose with an r.m.s.d. of 6.6 Å, yet the respective stems in the two forms are similar and superpose with r.m.s.d. values of 0.9, 2.6, and 1.4 Å for stems IIA, IIB, and IIC, respectively (Table S1).

### Opening of the three-way junction widens the major groove of the GGG motif by rearrangement of stems IIB and IIC

The closed and open forms of RRE SLII exhibit different conformations at the high-affinity Rev-binding site utilizing non-canonical base pairs. In the closed form, the GGG motif is locked into a dsRNA helix by the $G^8·U^{35}$ wobble and *syn*-$G^9·G^{34}$ purine–purine base pairs. In the open form, the GGG motif is more exposed by melting of the $G^8·U^{35}$ and *syn*-$G^9·G^{34}$ base pairs, and flipping of $G^8$ and $U^{35}$ out of the helix. As a consequence, the C1'–C1' distance between $G^8$ and $U^{35}$ increases from 12 Å in the closed conformation to 28 Å in the open conformation (Fig. 3A). The major groove formed at the GGG motif also widens from 12 Å (between P atoms of $G^8$ and $G^{30}$) in the closed conformer to 17 Å (between P atoms of $G^8$ and $G^{27}$) in the open conformer (Fig. 3A). In addition to widening of the major groove, loosening of these interactions in the open conformer increase the solvent accessible surface area of the three-way junction from 2350 to 2600 Å². Previous structural studies using stem IIB hairpin RNAs show that the major groove near the GGG motif widens from ~9 Å (P–P distance, PDB 1CSL) in the unbound form to ~15 Å (PDB 4PMI) in the bound form[46]. Thus, the open

conformation of RRE SLII with an enlarged major groove (~17 Å) and exposed surface area at the GGG motif suggests that Rev would selectively bind to the open conformation over the closed conformation (see Rev–SLII modeling section below).

### RRE SLII contains two Rev recognition sites, each of which can support Rev dimer binding

Rev consists of an ARM, a bipartite oligomerization domain (OD), and a nuclear export signal, separated by linker regions[29,47–52] (Fig. 4A). The α-helical ARM binds RNA and hosts the nuclear localization signal. The OD forms two α-helical regions and facilitates Rev–Rev dimer and oligomer formation. The C-terminal nuclear export signal is responsible for the recruitment of host nuclear export factors[33]. NMR and crystal structures of the stem IIB hairpin and Rev ARM (or N-terminal domain) complexes show that Rev ARM binds across the major groove of the RNA[46,53–56]. The Arg and Asn residues of ARM (R38, R39, N40, R41, R44, and R46) interact with the triple guanine motif by base-specific and non-specific interactions[46]. However, the stem IIB RNAs used in these structural and in vitro binding studies have an altered stem IIB sequence that stabilizes the GGG motif into an internal loop and are thus different from those observed in our native RRE SLII structures (Fig. S2B). Further, N40 of Rev, the major contributor to base-specific interactions with the GGG motif interacts differently in the crystal and NMR studies. In the crystal structure, N40 interacts across $G^9$ and $G^{34}$, whereas in the NMR structure it interacts with $G^9$ and $A^{35}$ despite both structures having the same sequence near the GGG motif[46,55]. Thus, the Rev interactions with the GGG motif seen in the previous structures may not represent native ARM recognition in SLII, and the GGG motif in native SLII may present different functional groups on the surface to facilitate novel base-specific Rev interactions.

To understand how the RNA structure mediates Rev oligomerization, we next investigated Rev interactions with RRE SLII using gel shift assays. Two Rev constructs were used to differentiate between protein–RNA and protein–protein interactions. First, a Rev construct with only the ARM ($Rev_{ARM}$) domain was used to identify RRE SLII recognition events (Figs. 4A and S3A). Second, a Rev construct limited to dimerization ($Rev_{70}$) was used to assess interaction between Rev monomers[22,57,58]. Rev contains a dimerization interface and a higher-order oligomerization interface. In $Rev_{70}$, L12S and L60R mutations were introduced within the higher-order oligomerization interface, thus limiting multimerization to dimer formation (Figs. 4A and S3A)[57].

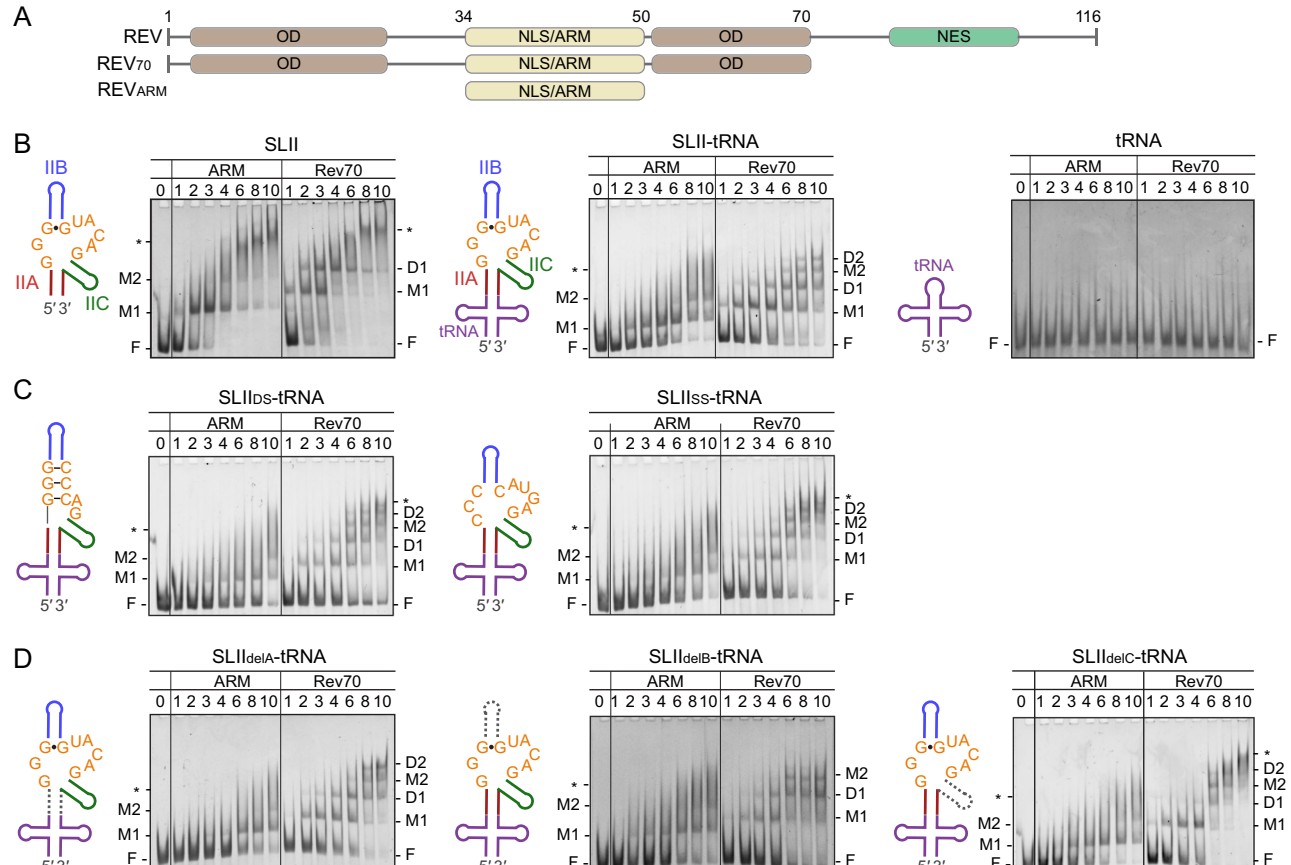

**Fig. 4 | Rev and RRE SLII binding assays. A** Schematics of Rev domains. Arginine-rich motif (ARM, beige) required for RNA interaction and oligomerization domains (OD, brown), facilitating oligomerization of Rev molecules are indicated. Rev_ARM was used to test RNA recognition and Rev_70 was used to assess dimer formation. **B** Interaction of SLII, SLII–tRNA, and tRNA with Rev proteins. Binding assays were performed with 500 nM RNA and increasing molar excess of Rev proteins. A schematic of the RNA used in the binding assay is shown to the left of the gel. SLII–tRNA is colored as in Fig. 1. The sequences of RNAs are listed in Fig. S4. SLII or SLII–tRNA contains two Rev_ARM binding sites, M1 and M2, representing a high-affinity and a low-affinity site. The two sites individually accommodate Rev_70

dimers, indicated as D1 and D2. Unbound RNA is denoted by F and higher-order oligomers are denoted with an asterisk (*). **C** Interaction of SLII three-way junction mutants with Rev proteins. SLII_DS–tRNA mimics the closed conformation by double-stranded C–G base pairs (left) and SLII_SS–tRNA mimics the ssRNA three-way junction structure (right). **D** Interaction of SLII stem deletion mutants with Rev proteins. SLII_delA–tRNA, SLII_delB–tRNA, and SLII_delC–tRNA mutants lack stems IIA, IIB, and IIC, respectively. The deleted stems are indicated with gray dotted lines in the RNA schematics. Each binding assay was performed three times with similar results. Uncropped gel images are provided in the Source Data file.

The binding assays show that RRE SLII has two Rev_ARM binding sites, high- and low-affinity sites. The first Rev_ARM binds to RRE SLII at an equimolar ratio of RNA at a high-affinity site (M1 in Fig. 4B, left) and the complex persists even at a tenfold excess of protein. The binding of Rev_ARM to a second, low-affinity site (M2 in Fig. 4B, left) is seen starting with a fourfold molar excess of protein to RNA, at which point there is no longer unbound SLII. The major Rev-bound species shift to higher-order complexes upon a sixfold excess of protein, appearing as a smear, suggesting non-specific Rev_ARM–RNA interactions. The initial binding of Rev_70 on SLII is similar to that of Rev_ARM. The first Rev_70 complex appears at the same molar concentration as the first Rev_ARM–SLII complex, suggesting that Rev_70 binds at the high-affinity binding site (M1 in Fig. 4B, left). However, the next Rev_70 and SLII complex forms much earlier than the binding of Rev_ARM to the low-affinity site, suggesting that two Rev_70 molecules dimerize at the high-affinity site (M1 and D1 in Fig. 4B, left). As the protein concentration increases, the Rev_70 dimer and RRE SLII complex shift into non-discernable higher-order oligomeric complexes.

Next, we studied the binding of Rev_ARM and Rev_70 to the SLII fused to the tRNA scaffold (SLII–tRNA). Rev_ARM and Rev_70 bind SLII–tRNA with a similar stepwise binding pattern as SLII (Figs. 4B, middle, and S3B). A single Rev_ARM binding to the high-affinity site is the

predominant RNP species up to a sixfold molar excess of protein, at which point binding of the second Rev_ARM to the low-affinity site occurs (M1 and M2 in Fig. 4B, middle). In the case of Rev_70, the monomer and dimer complexes on the high-affinity site are formed at similar protein concentrations as Rev_70–SLII complexes (M1 and D1 in Fig. 4B, middle). Compared to SLII interactions, additional Rev_70 monomer and dimer complexes at the low-affinity site were clearly visible at a sixfold molar excess of protein (M2 and D2 in Fig. 4B, middle). Rev does not interact with the tRNA scaffold, since binding of both Rev_ARM and Rev_70 to tRNA alone is minimal even at a tenfold excess of protein (Fig. 4B, right).

Taken together, binding assays with monomeric Rev_ARM and dimer-limited Rev_70 show that SLII has two Rev ARM binding sites, the high- and low-affinity sites. Following initial Rev ARM binding, the Rev_70 dimer assembles independently for each site, suggesting that Rev dimer formation is driven by protein–protein interactions between Rev monomers. Interestingly, the Rev and SLII–tRNA complexes show more discrete bands than SLII alone, suggesting that the tRNA scaffold prevents aberrant oligomer formation on RRE SLII by providing structural stability. The fusion of SLII to the tRNA scaffold mimics the connectivity of SLII to the rest of the full-length RRE (Fig. 1A, B). Since the stem-loops III–V of RRE do not contribute to Rev

binding and RRE activity directly[24], they may function similarly to the tRNA scaffold, providing structural stability to the Rev-binding sites in RRE (stems I and II) to assemble the functional RNP.

### Recognition of the initial Rev-binding site within the three-way junction requires both sequence and structural motifs

In the high-affinity purine-rich Rev-binding site, the triple guanine motif ($^8GGG^{10}$) and $^{34}GUACAG^{39}$ make up the three-way junction and are responsible for the closed and open conformations. To investigate how the conformation of the three-way junction affects Rev-binding, binding assays were performed with SLII constructs, in which mutations at the three-way junction were introduced to mimic the closed and open SLII conformations. Since the tRNA scaffold provides stability to the SLII structure and allows better resolution of the resulting complexes, mutations were introduced in SLII–tRNA. To mimic the closed SLII conformer, the $^8GGG^{10}/^{34}GUAC^{37}$ sequence was substituted with GGG/CCC such that the triple guanine motif is incorporated into dsRNA helix ($SLII_{DS}$), thus simulating closure by the $G^8 \cdot U^{35}$ and $G^9 \cdot G^{34}$ base pairs (Figs. 4C, left, and S3B). Binding assays with $SLII_{DS}$–tRNA show that the first $Rev_{ARM}$ and $Rev_{70}$ bind the RNA with reduced binding efficiency than WT SLII–tRNA (Fig. 4C, left). Thus, the initial recognition of the high affinity, triple guanine motif by $Rev_{ARM}$ is impaired by the closure of the SLII three-way junction, indicating that the GGG motif in the open conformation is required for primary Rev ARM binding. However, dimerization of $Rev_{70}$ on the RNA was not affected (Fig. 4C, left), suggesting that Rev dimerization is independent of recognition of the triple guanine motif [37,57,59]. Similarly, $Rev_{ARM}$ binding to the low-affinity site is also delayed, yet dimerization of $Rev_{70}$ at the site is comparable to the WT.

Next, the $^8GGG^{10}/^{34}GUAC^{37}$ sequence was substituted with CCC/CAUG ($SLII_{SS}$). Elimination of the triple guanine motif prevents $G^9 \cdot G^{34}$ and $G^8 \cdot U^{35}$-mediated closure of the SLII structure, forming single-stranded RNA as in the open conformation (Figs. 4C, right, and S3B). As expected, $Rev_{ARM}$ has a weak binding affinity for $SLII_{SS}$–tRNA, leading to delayed Rev binding on the initial binding site. Binding of a second $Rev_{ARM}$ is also delayed, and a discrete band is visible at an eightfold molar excess of protein. $Rev_{70}$ dimerizes efficiently at both sites, and $Rev_{70}$ complexes quickly shift into non-specific, higher-order complexes. Thus, the closure of the three-way junction or elimination of the GGG motif interferes with specific $Rev_{ARM}$ recognition of the initial binding site but does not affect subsequent $Rev_{70}$ dimerization. It was shown that substitutions of the GGG motif to CCC in RRE abolished Rev activity in cells[43]. Thus, although Rev interacts with SLII mutants and forms multiple RNP complexes, the resulting Rev oligomer and SLII complexes are not correctly configured for a functional RNP to recruit downstream nuclear export factors.

### Rev dimerization at the high- and low-affinity sites of RRE SLII requires stems IIC and IIB, respectively

Electrophoretic mobility shift assays show that $Rev_{70}$ can bind at least four sites on RRE SLII (Fig. 4B). Since the multimerization of Rev molecules requires protein–RNA as well as protein–protein interactions, unknown Rev-binding sites within RRE SLII branches may exist. To determine the Rev-binding sites in RRE SLII, binding assays were performed with SLII–tRNA mutants in which stems IIA, IIB, or IIC were deleted while preserving the initial high-affinity Rev-binding site (Figs. 4D and S3B). Deletion of stems IIA, IIB, and IIC ($SLII_{delA}$, $SLII_{delB}$, $SLII_{delC}$) resulted in similar $Rev_{ARM}$ binding phenotypes to WT SLII–tRNA with two bands (Fig. 4D), indicating that the Rev recognition site within the three-way junction of RRE SLII is likely intact.

Binding of $Rev_{70}$ to stem deletion constructs, however, shows different patterns of complex formation. Binding of $Rev_{70}$ to the $SLII_{delA}$–tRNA was similar to that of the WT SLII–tRNA, suggesting that specific interaction with stem IIA is not required in stabilizing Rev multimers (Fig. 4C, left). In contrast, $Rev_{70}$ binding assays with

$SLII_{delB}$–tRNA and $SLII_{delC}$–tRNA suggest that stems IIB and IIC are necessary for coordinating Rev dimerization. Deletion of stem IIB does not affect initial $Rev_{70}$ binding or dimerization at the high-affinity site. Additional $Rev_{70}$ binding to the low-affinity site is also similar to the WT SLII–tRNA (Fig. 4D, middle). However, $Rev_{70}$ does not dimerize on the low-affinity site even at a tenfold molar excess of protein, indicating that $Rev_{70}$ dimerization on the low-affinity site requires stem IIB. In contrast, the deletion of stem IIC inhibits dimerization of $Rev_{70}$ within the high-affinity site, as the progression from $Rev_{70}$ monomer to dimer-bound $SLII_{delC}$ species is stunted up to a fourfold molar excess of $Rev_{70}$ (Fig. 4D, right). As the protein concentration increases, the formation of Rev-bound species at the low-affinity site ensues. This suggests that the binding of the second Rev molecule at the high-affinity site requires interaction with stem IIC. Furthermore, the absence of stem IIC leads to the formation of non-specific higher-order complexes at high protein concentrations, in contrast to $Rev_{70}$ binding on WT SLII–tRNA or $SLII_{delB}$–tRNA (i.e., discrete band formation). Thus, stem IIC seems to provide structural stability to SLII to prevent non-specific Rev interactions with RNA.

Taken together, our Rev-binding assays show that SLII stabilized by the tRNA scaffold can assemble four Rev molecules as two dimers in discrete steps. The initial Rev binds at the high-affinity site (GGG site) via specific Rev ARM and RNA interaction. The second Rev then rapidly dimerizes with the first via protein–protein interactions between Rev OD. The second Rev likely binds in the major groove of the adjoining stem IIC. As the concentration of Rev protein increases, a third Rev binds at a second, low-affinity site near the three-way junction of RRE SLII, which is distinct from the initial Rev-binding site. Finally, the fourth Rev dimerizes with the third Rev molecule likely in stem IIB. Notably, the assembly of Rev on the $SLII_{delB}$ mutant is halted at three Rev proteins even at a tenfold molar excess of protein, suggesting that the RNA structure can control the sequential Rev assembly on RRE. We propose that the conformational states of RRE SLII may act as a checkpoint to regulate its recognition by Rev (see "Discussion").

## Discussion

### Model of Rev dimerization on the initial high-affinity site within the full-length RRE SLII

During HIV infection, partially spliced and unspliced viral RNAs need to be exported from the host cell nucleus to the cytoplasm to synthesize viral proteins and assemble virions. To mediate nuclear export, HIV uses specific interaction between Rev protein and an intronic RNA sequence RRE. Multimeric Rev proteins bind to RRE and recruit the nuclear export factors for cytoplasmic translocation of the viral RNAs. However, the architecture of the highly specific RRE-Rev multimeric complex is unknown, largely due to the lack of an RRE structure. In the structure of a $Rev_{70}$ dimer in complex with a modified stem IIB hairpin RNA (PDB code 4PMI), one Rev monomer interacts with the GGG sequence within the widened major groove and a second Rev monomer binds non-specifically along the adjacent major groove[46]. Our RRE SLII structure now shows that the purine-rich, high-affinity Rev-binding site (the GGG motif) is incorporated into the three-way junction rather than the predicted internal loop in stem IIB. Thus, the specific Rev–RRE SLII interactions are likely different from those reported in the Rev and stem IIB hairpin complexes (Fig. S3).

To provide insight into RNA-directed Rev dimerization on RRE, we modeled a Rev dimer on RRE SLII. Comparison of the closed and open structures suggests that pivoting of stems IIB and IIC at the three-way junction in the open conformation increases the width of the major groove at the high-affinity Rev-binding site by ~5 Å (Fig. 3A). Since Rev binding requires a widened major groove of the RNA, we chose the open conformer of SLII for modeling the SLII–Rev dimer complex. Guided by the Rev dimer and stem IIB hairpin complex structure[46], the first Rev molecule was placed in the major groove formed by the GGG motif. Rev monomer fits well in the open form of SLII within the

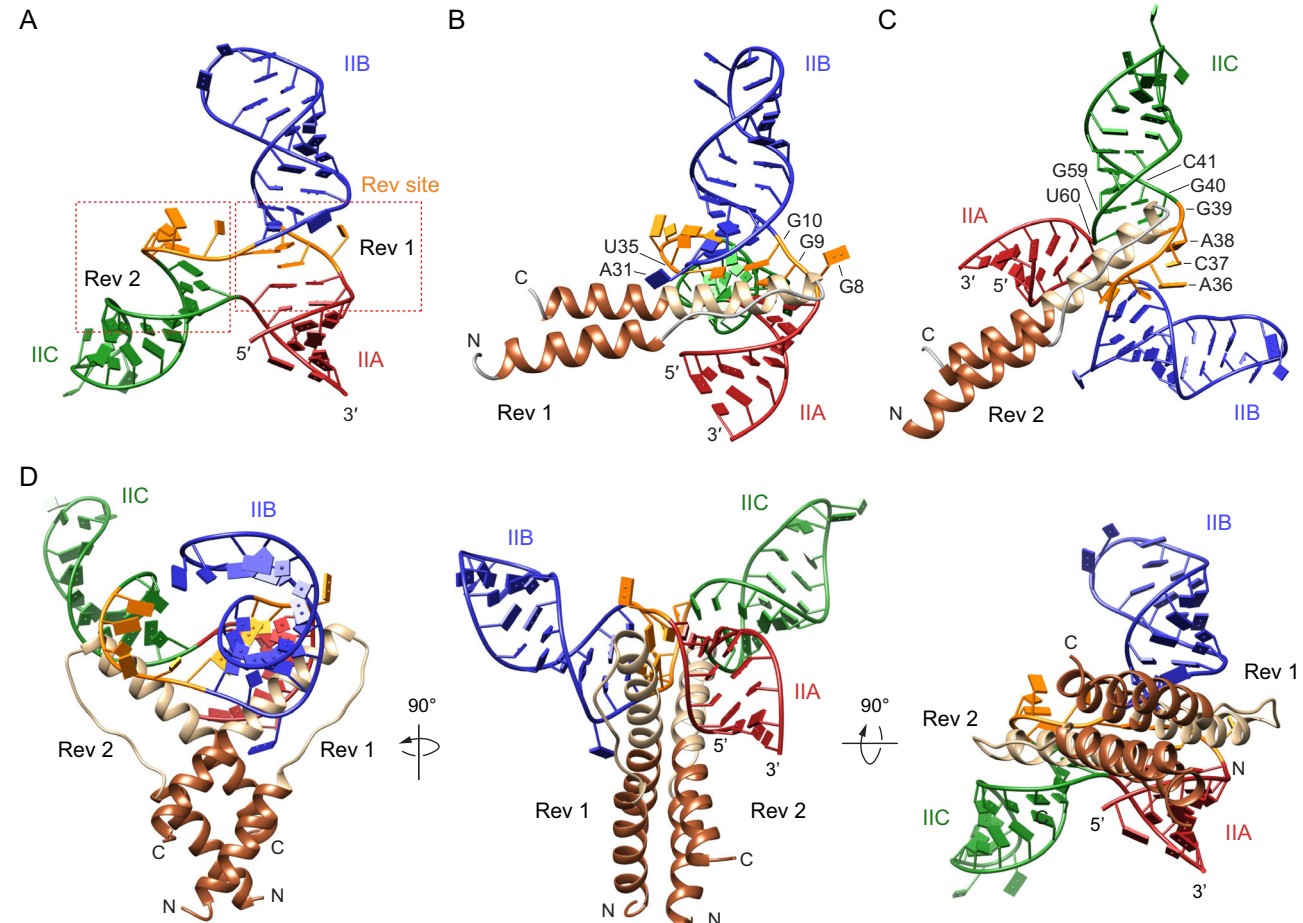

**Fig. 5 | Model of Rev dimer binding onto the open conformation of RRE SLII.**
**A** Rev dimer binding site in RRE SLII. The first Rev binds in the high-affinity binding site, and the second Rev likely binds in the major groove next to the high-affinity site. The two Rev-binding sites are indicated with red boxes. **B** First Rev binding on the initial high-affinity binding site. Rev arginine-rich motif (ARM) and two oligo-merization domains (OD) are shown in beige and brown, respectively. The ARM of

the first Rev molecule was placed within the widened major groove formed by [8]GGG[10] of the open SLII conformer. **C** Second Rev molecule binding within stem IIC. The ARM of the second Rev molecule was placed within the major groove at the base of stem IIC. **D** Binding of the Rev dimer on the open SLII conformation. The Rev dimer is stabilized by the interaction of the ARM domains within the RNA major grooves as well as between neighboring Rev ODs.

widened major groove of the three-way junction without major steric clash (Fig. 5A, B). In this orientation of the Rev monomer, the ARM domain would interact with $G^9$ and $G^{10}$, *syn*-$G^{34}$, and the flipped $A^{31}$ bulge nucleotide. By virtue of dimerization, the second Rev monomer is poised to interact with the major groove of the adjacent stem-loop IIC along the three-way junction (Fig. 5C, D). The N-terminal domain of Rev ($Rev_{70}$) has been crystallized in various dimer forms, where the angle between the two ARM peptides ranges from 40 to 140°, sug-gesting that Rev may accommodate differently arranged RNA structures[46,60,61]. Thus, to fit the Rev dimer into the three-way junction of RRE SLII, the angle between the two ARMS of the Rev dimer was increased from 50° (in the IIB hairpin complex) to 115° (Fig. 5C).

In the SLII−Rev dimer model, the Rev dimer sits astride the three-way junction with each monomer binding along the major grooves toward stems IIB and IIC (Fig. 5D). The first Rev monomer binds in the high-affinity site in the three-way junction and dimerizes with the second Rev monomer that binds in the major groove toward stem IIC. This arrangement of the Rev dimer is consistent with previous Rev-binding studies where in the presence of Rev, RRE showed reduced SHAPE activity for the nucleotides in the three-way junction ($G^9$, $G^{33}$, $A^{36}$, and $A^{38}$; SLII numbering)[62]. Further, our Rev-binding assays show that the initial, high-affinity Rev dimer is assembled at the three-way junction and adjoining stem IIC (Fig. 4D). In contrast to the enlarged major groove of the first Rev-binding site, the width of the major

groove in the second Rev-binding site is ~10 Å in both SLII conforma-tions (Fig. 5A). Thus, binding of a second Rev would require widening of the major groove in the second Rev-binding site. It is possible that the binding of the first Rev molecule in the high-affinity site changes the conformation of the second Rev-binding site, making it more conducive for Rev binding. The $G^{40}$·$U^{60}$ wobble pair at the foot of stem IIC may facilitate the widening of the major groove to allow binding of the second Rev. Previously, an RRE-Rev binding model was proposed based on small-angle X-ray scattering (SAXS) data of RRE[18]. Because the spacing (~55 Å) between the stems II and I in the "A"-shaped RRE model is similar to the spacing between the two RNA-binding domains in Rev dimers, an initial Rev dimer was proposed to bridge the stems II and I Rev-binding sites (IIB and IA sites)[18]. Our Rev-binding model suggests that SLII alone binds the initial Rev dimer, and is thus different from the SAXS-based model.

### RRE modulates the degree of Rev binding via progressive con-formational changes

RRE contains multiple Rev-binding sites, yet only two sequence-specific Rev-binding sites have been identified, one in the stem II junction and the other in stem IA, suggesting that RRE utilizes largely non-sequence-specific interactions to form the RNP complex. Addi-tional Rev-binding sites on RRE are thus likely created sequentially through structural rearrangement of RRE as new Rev molecules

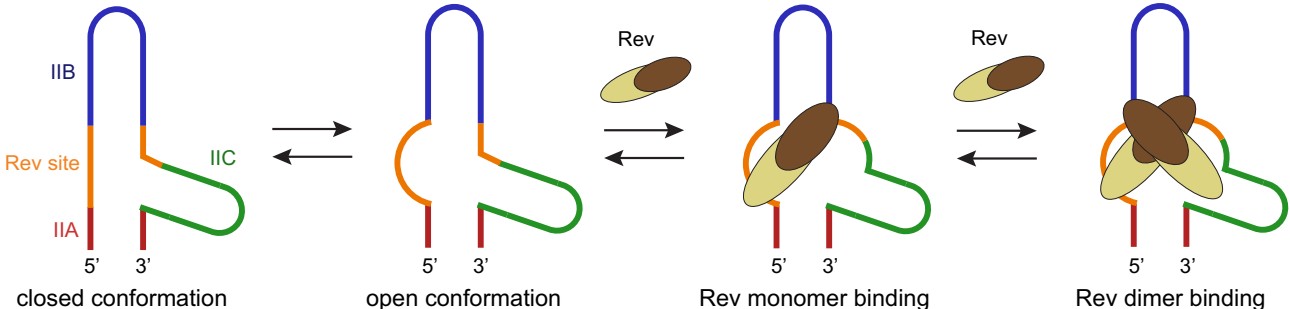

**Fig. 6 | Model of progressive Rev binding on RRE SLII.** The closed and open RRE SLII exist in an equilibrium. The conformational changes between the closed and open SLII depict the first step in Rev oligomerization: recognition and binding of the high-affinity Rev-binding site. A single Rev molecule binds the open SLII conformer at the high-affinity binding site and induces a conformation change in the nearby major groove at stem-loop IIC. This allows a second Rev monomer to dimerize on the bound Rev molecule via interaction with stem IIC. The binding of additional Rev molecules continues to induce conformational changes at nearby sites on RRE until all Rev-binding sites are occupied.

interact. The native full-length SLII exists in a dynamic equilibrium of two alternative conformations, closed and open, mediated by non-canonical base pairing in the three-way junction. We propose that the closed and open states have low- and high-affinity for Rev, respectively, and that Rev preferentially binds to the open conformation (Fig. 6). Binding of the first Rev molecule then changes the conformation of the nearby Rev-binding site on RRE and increases its binding affinity for a second Rev molecule (positive cooperativity). Similar sequential conformational changes upon Rev binding to a nearby site will continue until all Rev-binding sites are occupied. This cooperative binding model suggests that RRE conformational states regulate the binding of Rev molecules in a stepwise fashion by modulating RNA-binding affinity. Previously, single-molecule FRET studies of RRE and Rev in the presence of DEAD-box helicase DDX1 showed that DDX1 accelerates the initial Rev interaction with RRE[63]. It was proposed that DDX1 could act as an RNA folding chaperon and induce an altered RRE RNA structure that has a higher affinity for Rev. Our Rev-binding model facilitated by progressive conformational changes in RRE would be consistent with the model, where DDX1 facilitates a conformational change from the closed to open state of RRE, thus accelerating the initial Rev interaction with RRE. We show that Rev oligomerization on RRE SLII is a controlled process. Native RRE and Rev-bound RRE species exist in equilibrium and assemble discrete RNA–protein complexes, rather than exhibiting a full shift into higher-order oligomeric complexes (Figs. 4B and 6). This highly specific initial Rev binding and subsequent cooperative multimerization of Rev likely allows RRE to function as a rheostat for Rev concentrations.

Regulation of the multimeric Rev–RRE complex acts as a biological timer for the formation of the nuclear export-competent RNP complex and thus transition into late-stage HIV viral infection. The importance of this regulatory axis is demonstrated in HIV-positive individuals, in which more active Rev and RRE isolates were associated with individuals with advanced disease states[64,65]. Due to the flexibility of RNA secondary structures, mutations within the RRE may provide more immediate modulation of the Rev–RRE axis to upregulate nuclear export under immunocompromised instances[19]. Since the export of RRE-containing RNAs is essential for HIV replication, the RRE and Rev complex is an attractive therapeutic target to trap HIV in a latent phase. The structure of RRE SLII would be useful to develop small molecules that bind at the three-way junction and inhibit the conformational changes required for cooperative binding of Rev molecules and the formation of the functional Rev–RRE complex.

## Methods
### Design and purification of tRNA-scaffolded SLII RNA constructs
The SLII–tRNA fusion constructs were designed in a pBlueScript II SK(+) vector as previously described[39,40,66]. Briefly, the stem-loop II

sequence of HIV-1 RRE (nt 7353–7418, GenBank: AF033819.3) was inserted into the anticodon loop of a human lysine tRNA to generate the RRE SLII–tRNA chimera. Stem IIA is continuous with the anticodon stem of tRNA. An additional G54 was added to generate a stable tetraloop ($^{51}$AAUG$^{54}$) in stem IIB. To test interaction with Rev proteins, deletion and substitution mutants, SLII$_{DS}$, SLII$_{SS}$, SLII$_{delA}$, SLII$_{delB}$, and SLII$_{delC}$ with the tRNA scaffold were designed (Table S1) and generated using site-directed mutagenesis of SLII–tRNA. The secondary structure prediction using the RNAfold server within the ViennaRNA Web Services (http://rna.tbi.univie.ac.at/)[67] indicated that the modified RNA constructs maintain the tRNA structure.

The tRNA fusion constructs were purified as previously described[39]. The plasmids were freshly transformed into BL21 *E. coli* cells and grown overnight in 2 L of 2× TY medium with 100 μg/mL ampicillin at 37 °C. The cells were harvested at 4000 × *g* for 30 min and resuspended in 10 mL of buffer containing 10 mM magnesium acetate and 10 mM Tris-HCl, pH 7.4. The RNA was extracted by adding 12 mL of acid phenol:chloroform (5:1) mixture and gently agitated for 1 h at room temperature. The solution was next centrifuged at 21,000 × *g* for 30 min at 20 °C. The aqueous phase was extracted and mixed with 0.1 volume of 5 M NaCl and 2 volumes of absolute ethanol for RNA precipitation. Following incubation at −20 °C overnight, the RNA was pelleted by centrifugation at 21,000 × *g* for 30 min at 4 °C, dissolved in buffer A (40 mM sodium phosphate, pH 7.0), and passed through a 0.22 μm filter. The RNA was loaded onto a HiLoad 16/20 Q ion-exchange column (Cytiva, Marlborough, MA). The desired RNA eluted between 520 and 560 mM NaCl in buffer A. The fractions were analyzed on a urea–acrylamide gel. RNA-containing fractions were then pooled and buffer-exchanged into water. Typical yield of RNA is 30–40 mg from a 2-L culture.

### In vitro transcription and purification of SLII and tRNA control
The DNA templates for SLII and tRNA$^{Lys}$ were individually generated using PCR from the SLII–tRNA plasmid. RNA was in vitro transcribed using standard T7 RNA polymerase transcription methods[68]. The transcription reaction was prepared by combining 10 mM rNTPs, 20 nM DNA template, and 200 μg/mL T7 RNA polymerase[69] in buffer containing 40 mM Tris-HCl, pH 8.3, 20 mM MgCl$_2$, 2.5 mM spermidine, 10 mM DTT, and 0.01% Triton X-100. In vitro transcribed RNA was diluted in buffer A (40 mM sodium phosphate, pH 7.0) and loaded onto a MonoQ 5/50 GL ion-exchange column (Cytiva, Marlborough, MA). The desired RNA eluted between 560 and 600 mM NaCl in buffer A. Fractions were analyzed on a urea–polyacrylamide gel, and RNA-containing fractions were pooled and buffer-exchanged into water. Typical yield is ~20 μg RNA from 1 mL T7 RNA transcription reaction.

## Expression and purification of Rev proteins

The design of the HIV Rev$_{70}$ protein is as previously described[22,70]. Rev$_{70}$ includes only the ordered N-terminal domain (NTD, amino acids 1–70, UniProt ID: P69718). Rev$_{70}$ was designed as a GB1 (B1 domain of Streptococcal Protein G)-fusion protein within the pET-28 vector and contains E47A, L12S, and L60R mutations, the latter two of which limit Rev higher-order oligomerization to dimer formation[57,60]. Rev$_{ARM}$ (amino acids 34–50) was generated using site-directed mutagenesis of the Rev$_{70}$ plasmid. The entire sequences of the GB1–Rev$_{70}$ and GB1–Rev$_{ARM}$ constructs are listed in the Source Data file. To purify Rev$_{70}$, the plasmid was transformed into *E. coli* BL21(DE3) cells, and cells were grown overnight at 37 °C in 1 L of TB medium with 30 μg/mL of kanamycin. Protein expression was induced with 1 mM IPTG for 4 h at 37 °C. The cells were harvested at $4000 \times g$ and resuspended in lysis buffer containing 50 mM Tris-HCl, pH 7.4, 500 mM NaCl, 100 mM $(NH_4)_2SO_4$, 0.1% Tween20, and cOmplete EDTA-free Protease Inhibitor Cocktail (Roche, Indianapolis, IN). The cells were lysed by sonication and centrifuged at $21,000 \times g$ for 30 min, and the supernatant was collected. The protein was purified using TALON immobilized metal affinity chromatography resin (Takara, San Jose, CA) equilibrated with 40 mM Tris-HCl, pH 7.4, 500 mM NaCl, and 5 mM imidazole. Protein was eluted using a gradient of 5–200 mM imidazole, and fractions were analyzed using a 15% SDS–polyacrylamide gel. Fractions containing protein were concentrated and dialyzed into a buffer containing 40 mM Tris-HCl, pH 7.4, 250 mM NaCl, and 400 mM $(NH_4)_2SO_4$. Rev$_{ARM}$ was similarly expressed and purified except that all buffers do not contain $(NH_4)_2SO_4$.

## Crystallization, X-ray data collection, and structure determination

For crystallization, SLII–tRNA was reconstituted at 20 mg/mL in 10 mM Tris pH 7.0, 50 mM NaCl, and 10 mM MgCl$_2$. Initial crystals of SLII–tRNA appeared in the PACT high throughput screen condition F2, containing 200 mM sodium bromide, 100 mM Bis-Tris propane pH 6.5, and 20% polyethylene glycol (PEG) 3350. Following optimization of the condition, diffraction quality crystals were obtained using the hanging-drop vapor diffusion method at 18 °C by mixing the RNA with an equal volume of reservoir solution containing 100 mM S.P.G. (buffer mixture of succinic acid, sodium dihydrogen phosphate, and glycine in a 2:7:7 molar ratio) pH 7.0, 20–24% PEG-6000, and 200 mM sodium fluoride[71]. Prior to sealing the drops, acetone was added to 8% into the reservoir solution, as previously described[40]. Crystals appeared within 3 days and grew to full size within 2 weeks. For data collection, crystals were passed through mother liquor with 20% (v/v) glycerol, cryoprotected in paratone-N oil, and flash frozen in liquid nitrogen. Diffraction data to 2.85 Å resolution were collected at the Advanced Light Source beamline 8.2.2 (Lawrence Berkley National Laboratory, California). The dataset was processed using XDS with anisotropic correction[72]. The crystal belonged to space group P2$_1$ with unit cell dimensions of $a = 67.6$ Å, $b = 81.2$ Å, $c = 81.1$ Å, and $\beta = 99.1°$, and contained two molecules in the asymmetric unit. The initial solutions were found by molecular replacement with tRNA (PDB 7LYG) and small dsRNA helix (PDB 3ND3) as search models using the Phaser program within the Phenix application suite[73]. The final model contained two SLII–tRNA molecules with R and R$_{free}$ of 0.23 and 0.26, respectively. Atomic coordinates for SLII–tRNA are deposited to PDB with accession number 8UO6. For structural analysis, chimera was used to determine atomic distances, angles, and surface areas[74].

## Modeling of Rev dimer on RRE SLII

The crystal structure of the Rev dimer and RRE stem-loop IIB hairpin complex (PDB 4PMI) was used to model Rev binding onto the open RRE SLII conformer. First, the Rev–RRE stem-loop IIB complex was superposed with the open RRE SLII conformer by aligning the GGG motif and stem IIB hairpin. This places the first Rev molecule within the high-affinity site at the three-way junction of the open SLII conformer. The position of the first Rev molecules was manually adjusted to better fit into the widened major groove formed at the GGG motif of SLII. Placement of the first Rev molecule orients the ARM of the second Rev molecule of the dimer toward stem IIC. Since the angle between the two ARMs of Rev dimer was shown to range between 40 and 140°[46,60,61], the angle between the two ARMs was adjusted to place the second Rev molecule into the major groove at the base of stem IIC.

## RNA–protein binding gel shift assays

Electrophoretic mobility shift assays were used to measure the interaction between Rev$_{70}$ or Rev$_{ARM}$ with native SLII and tRNA fused SLII constructs, SLII, SLII$_{DS}$, SLII$_{SS}$, SLII$_{delA}$, SLII$_{delB}$, and SLII$_{delC}$. tRNA$^{Lys}$ was also included as a negative control. Both Rev$_{ARM}$ and Rev$_{70}$ have the GB1 linker to allow for better resolution of distinct RNP species. RNA (500 nM) was incubated with increasing molar amounts of Rev protein (500 nM to 5 μM) in binding buffer (20 mM Tris-HCl, pH 7.4, 100 mM NaCl, and 10 mM MgCl$_2$) for 15 min at room temperature and loaded on an 8% native polyacrylamide gel. The gel was stained with ethidium bromide to visualize RNA. The binding assays were performed in triplicate with similar results. Uncropped gel images are provided in the Source Data file.

## Reporting summary

Further information on research design is available in the Nature Portfolio Reporting Summary linked to this article.

## Data availability

The data supporting the findings of this study are available from the corresponding author upon request. Atomic coordinates and structure factors for SLII–tRNA have been deposited in the RCSB Protein Data Bank under the accession code 8UO6. Accession codes with hyperlinks used in this manuscript are as follows: PDB 7LYG, PDB 3ND3, PDB 1CSL, PDB 4PMI, UniProt P69718, GenBank AF033819.3 [https://www.ncbi.nlm.nih.gov/nuccore/AF033819]. Source data are provided with this paper.

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

## Acknowledgements

We thank Dr. Mark White of Sealy Center for Structure Biology at UTMB, Juan Nysschen of IU Macromolecular Crystallography Facility, and Dr. Jay Nix at Advanced Light Source for help with X-ray data collection and helpful discussions. Beamline 8.2.2 of the Advanced Light Source, a DOE Office of Science User Facility under Contract No. DE-AC02-05CH11231, is supported in part by the ALS-ENABLE program funded by the National Institutes of Health, National Institute of General Medical Sciences, grant P30 GM124169-01. This research used resources of the Advanced Photon Source, a U.S. Department of Energy (DOE) Office of Science User Facility operated for the DOE Office of Science by Argonne National Laboratory under Contract No. DE-AC02-06CH11357. Use of the LS-CAT Sector 21 was supported by the Michigan Economic Development Corporation and the Michigan Technology Tri-Corridor (Grant 085P1000817). This work was supported by NIH grants R01AI187856 and U19AI171413 (to K.H.C.), Jeane B. Kempner pre-doctoral fellowship (to J.T.) and Institute for Human Infections and Immunity (IHII) Data acquisition grant (to K.G.).

## Author contributions

K.H.C., K.G. and J.T. conceived the study. J.T. and M.S. prepared various RNAs and crystallized. J.T. collected X-ray diffraction data, and J.T., G.G. and K.H.C. processed the data. J.T. determined the crystal structure. J.T. and K.G. prepared Rev proteins and J.T. performed binding assays. J.T., K.G. and K.H.C. analyzed the structure and wrote the manuscript.

## Competing interests

The authors declare no competing interests.
