## [Peer Review File · Nature Communications]

Structure of HIV-1 RRE stem-loop II identifies two conformational states of the high-affinity Rev binding siteREVIEWER COMMENTS

Reviewer #1 (Remarks to the Author):

This paper describes the crystal structure of the HIV SLII-tRNA determined in two conformations. Binding studies and molecular modeling were also used to validate the crystal structure and derive a model for Rev recognition of the RRE. Based on all evidence and available literature, the authors propose a model for Rev recognition of RNA that fills gaps and inconsistencies in the literature. Overall, I find the paper to be excellent. I think the model is sound and agrees with the existing literature. Rev is made in just a few copies in infected cells. So, the idea that Rev's initial binding to RNA is specific, followed by subsequent cooperative multimerization (aka a 'rheostat for Rev concentrations'), makes perfect sense.

I only have a few minor comments that I invite the authors to address:

- 1 - Crystallographic Table. Include refinement statistics for reflections in the outer shell
- 2 - Page 12 "However, dimerization of Rev70 on the RNA was not affected, suggesting that Rev dimerization is independent of recognition of the triple guanine motif (Fig 4C, left)." Provide a reference for this statement.
- 3 - SLII-tRNA as crystallized using 200 mM sodium fluoride. Can the authors clarify what it's for? Can they see the density of fluoride? Are there unassigned peaks of density? What happens if omitted?
- 4 - Have the authors tried to add sodium fluoride to a bandshift assay to see if it can change the pattern of intermediates?
- 5 - Can the authors model and explain the putative role of Phosphorylation in Rev? Rev is phosphorylated for most of its lifespan in human cells.
- 7 - Most bandshifts in this paper were carried out for Rev 1-70. What would happen if the full-length protein was used? Is the C-terminus doing anything as far as RNA binding?

Reviewer #2 (Remarks to the Author):

The manuscript by Tipo et al. presents an interesting RNA crystal structure of a larger segment of the Rev Responsive Element RNA that reveals two different conformations. The crystallography analysis and Rev binding assays are carefully done. The closed conformation includes a G8-U35 wobble and syn G9-G34 pair that changes conformation in the open conformation. Interestingly, the G10 is also in the syn conformation and stacks on the syn G9. The syn G conformations are very interesting and will provide important benchmarks for computational studies aimed at understanding and predicting RNA structures. The manuscript also presents a model of how the open conformation might interact with the Rev protein and how the dynamic changes in RNA may regulate Rev binding. The results and proposed model do raise some important questions that need to be addressed.

1. The several guanine nucleotides change conformation from syn to anti between the open and closed conformations. In the closed conformation, G9 and G10 adopt the anti conformation, and G34 adopts the syn conformation while forming a pair with G10. What would provide the energy to overcome the high energy barrier to these conformational changes?
2. In the open conformation, the G8 nucleotide flips out of the helix (Figure 2D). Are there new contacts that form to stabilize this conformation of G8 or does the G8 nucleotide become highly flexible? The crystal lattice contacts and examples of the electron density should be included in figures in supporting information.
3. The authors could provide more comparisons and discussion of previous models for Rev-RRE interactions based on biophysical studies and smFRET. For example, *Nucleic Acids Research* v 45 p 4632 and *J. Molecular Biology* v 430 p 537 propose an alternative model that invokes additional helicase protein interactions. Are crystal structures presented in this manuscript consistent or different than previously proposed models for Rev-RRE interactions?

Reviewer #3 (Remarks to the Author):

This manuscript describes the elucidation of the 3D structure of the entire HIV-1 Rev-response element (RRE) stem-loop II for the first time; previous studies used truncated constructs or generated a low resolution SAXS model with a significantly longer construct. The tRNA scaffold approach previously developed by the authors, as well as the introduction of a single G nucleotide to form a stable tetraloop, were likely key in generating crystals that diffracted to reasonably high (2.85 Å) resolution. The RNA structure determined in this new work reveals distinct closed and open conformations. Surprisingly, the high-affinity Rev-binding site is located within the three-way junction rather than the predicted stem IIB. The data generated both by X-ray crystallography and gel shift binding assays are convincing. Only the open conformation has a significant widening of the major groove capable of accommodating Rev interaction. Rev binding assays show that RRE stem-loop II has high- and low-affinity binding sites, each of which binds a Rev dimer. The authors propose an overall binding model, wherein Rev-binding sites on RRE are sequentially created through structural rearrangements induced by Rev-RRE interactions.

The results presented in this manuscript, which is generally clearly written, represent a significant advance in the field. A surprising finding was the lack of a predicted internal loop in the stem IIB (the high-affinity Rev binding site) and a much longer than predicted three-way junction. Instead of the internal loop structure, the high-affinity 8GGG10 motif is part of a dsRNA stem in the closed conformation and partially single-stranded in the open conformation. The two conformations identified by this work also help to explain previous discrepancies in the field based on structure-probing data.

Major comments

1. Figures: avoid use of red/green and other color-blind inaccessible colors.
2. Figure 2: what do the filled vs open boxes around the nt in panels B and D. Are these syn vs. anti nt and if so, the designation in D is not correct?
3. Legend to Figure 4: this is very long and could be significantly reduced to only describe the figure without interpreting the results, which is redundant with the text.
4. Is the final model consistent with the model for the full RRE generated using SAXS (2013)? I'm not sure if this comparison is valid but a comment for those familiar with that over all letter "A" form structure would be helpful.

Minor comments

1. Text requires careful proofreading for some writing issues such as missing words (e.g., Abstract, 3rd line up, "Based on these results and...."; Intro, 4 lines down "...the early stage of HIV infection,..."), failing to define some terms (nt, WT, etc) and a few grammatical errors.
2. Page 4, what is meant by "primes" in the context of Rev recognition? This is a vague term and not standard in the field.
3. Page 4 also has an example of a paragraph that it too long and could be split following ref 38, starting a new paragraph with "To better understand...."
4. Throughout the manuscript, the authors use "in vivo" when in fact they mean "in cells" as these were not animal studies.
5. The authors show avoid "booster" words such as "drastically"
6. The apostrophes should be change to prime symbols (e.g. page 8 the C1'-C1' distance).
7. Methods: what is the source of T7 polymerase used? (should also be T7 RNA polymerase)

Dear Editor,

We thank the reviewers for reviewing our manuscript “Dual Conformational States of HIV-1 RRE Stem-Loop II: Implications for the Initial High-Affinity RRE-Rev interaction” by Tipo et al. We believe the revised manuscript is much improved and addresses reviewers’ concerns and comments. We list our responses to the reviewers’ comments below. If there are any further questions, please do not hesitate to contact us.

Reviewer #1:

1 – Crystallographic Table. Include refinement statistics for reflections in the outer shell.

We added refinement statistics in Table 1.

2 – Page 12 “However, dimerization of Rev70 on the RNA was not affected, suggesting that Rev dimerization is independent of recognition of the triple guanine motif (Fig 4C, left).” Provide a reference for this statement.

The statement (Rev dimerization is independent of recognition of the triple guanine motif) is based on our EMSA result, because two ARM peptides (that lack the oligomerization domain) bind RRE stem-loop II, while four Rev70 bind stem-loop II via both RNA-protein and protein-protein interactions. This observation is consistent with previous reports. We added references.

3 - SLII-tRNA as crystallized using 200 mM sodium fluoride. Can the authors clarify what it’s for? Can they see the density of fluoride? Are there unassigned peaks of density? What happens if omitted?

Initial crystallization conditions were obtained from the PACT high throughput screen F2 (0.2M sodium bromide, 0.1M Bis-Tris propane pH 6.5, and 20% PEG 3350). We thus screened for other halogen salts, and sodium fluoride yielded diffraction quality crystals. Although crystals grown without sodium fluoride (or any halogen salt) did not diffract well, we did not observe density for fluoride. We modified the text to include initial crystallization condition.

4 – Have the authors tried to add sodium fluoride to a band shift assay to see if it can change the pattern of intermediates?

We did not test the effect of sodium fluoride in EMSA, as the fluoride was presumed to stabilize the crystal lattice. When we tested sodium chloride concentrations in the binding conditions, no significant changes were observed from 0.1 to 0.5 M NaCl.

5 – Can the authors model and explain the putative role of Phosphorylation in Rev? Rev is phosphorylated for most of its lifespan in human cells.

Previous studies show that Rev is phosphorylated at Ser5, 8, 54, 56, and within the C-terminal 30 amino acids (Cochrane et al., JVI, 1989; Fouts et al., Biochemistry, 1997). Because Ser 5/8/54/56 are located outside the RNA binding domain, and Rev70 lacks the C-terminal 50 amino acids, our modeling and EMSA data would not be able to address the biological function of Rev phosphorylation.

7 – Most band shifts in this paper were carried out for Rev 1-70. What would happen if the full-length protein was used? Is the C-terminus doing anything as far as RNA binding?

The N-terminal 70 amino acids of Rev (Rev70) are shown to be sufficient for Rev binding and oligomerization onto RRE (Zapp et al., PNAS, 1991). The disordered C-terminal domain (containing the nuclear export signal) recruits the cellular host factors involved in nuclear export. As the C-terminal domain is not involved in RNA interaction or Rev oligomerization, we expect the same binding results as Rev70.

Reviewer #2 :

1. The several guanine nucleotides change conformation from syn to anti between the open and closed conformations. In the closed conformation, G9 and G10 adopt the anti conformation, and G34 adopts the syn conformation while forming a pair with G10. What would provide the energy to overcome the high energy barrier to these conformational changes?

Both open and closed structures likely reflect low energy conformations. Both conformations have a *syn-G:anti-G* base pair and an unpaired *syn-G*. Although *syn-G* is higher energy than *anti-G*, formation of the *syn-G:anti-G* base pair would minimize the steric clash from *anti-G: anti-G* base pair and could be stabilized by hydrogen bonding and stacking interactions. Although the *syn* conformation for free purines is slightly disfavored by ~ 1.2 kcal/mol in ΔG at 35°C, it has been found in a majority of functional RNAs with tertiary structures (Sokoloski et al., RNA 2011). Further, the open conformation could be stabilized by interaction with solvent molecules although they are not modeled in the crystal structure. It is likely that the energy needed to overcome the energy barrier in the conformational switch from the closed to open conformation is provided by the Rev binding as well as interaction with other host proteins such as DDX1.

2. In the open conformation, the G8 nucleotide flips out of the helix (Figure 2D). Are there new contacts that form to stabilize this conformation of G8 or does the G8 nucleotide become highly flexible? The crystal lattice contacts and examples of the electron density should be included in figures in supporting information.

The G8 nucleotide that flips out of the helix contains full density and is not stabilized via solvent interactions nor crystal contacts (see the figure on the right). Fig 2A and 2C show the density surrounding the RNA. In addition, we added a new supplementary figure S1 showing the crystal lattice contact.

3. The authors could provide more comparisons and discussion of previous models for Rev-RRE interactions based on biophysical studies and smFRET. For example, Nucleic Acids Research v 45 p 4632 and J. Molecular Biology v 430 p 537 propose an alternative model that invokes additional helicase protein interactions. Are crystal structures presented in this manuscript consistent or different than previously proposed models for Rev-RRE interactions?

The above-mentioned papers describe that the presence of DDX1 accelerates the initial Rev interaction with RRE, and propose a model where DDX1 acts as an RNA folding chaperon, producing an altered RNA structure that has a higher affinity for Rev. This model would be consistent with our sequential Rev binding model. We added the following sentences in Discussion.

“Previously, single molecule FRET studies of RRE and Rev in the presence of DEAD-box helicase DDX1 showed that DDX1 accelerates the initial Rev interaction with RRE (62). It was proposed that DDX1 could act as an RNA folding chaperon and induce an altered RRE RNA structure that has a higher affinity for Rev. Our Rev binding model facilitated by progressive conformational changes in RRE would be consistent with the model, where DDX1 facilitates a conformational change from the closed to open state of RRE, thus accelerating the initial Rev interaction with RRE.”

Reviewer #3:

1. Figures: avoid use of red/green and other color-blind inaccessible colors.

We checked the colors with color proofs in Adobe illustrator.

2. Figure 2: what do the filled vs open boxes around the nt in panels B and D. Are these syn vs. anti nt and if so, the designation in D is not correct?

The filled boxes indicate the nucleotides involved in the interactions mediating non-canonical base pairs and the opening/closing of the three-way junction. The same nucleotides are displayed in ball-and-stick figures to the right. We modified the figure legend to clarify.

3. Legend to Figure 4: this is very long and could be significantly reduced to only describe the figure without interpreting the results, which is redundant with the text.

We have shortened the figure legend.

4. Is the final model consistent with the model for the full RRE generated using SAXS (2013)? I'm not sure if this comparison is valid but a comment for those familiar with that over all letter "A" form structure would be helpful.

In the published SAXS model, the stem-loop II was modeled as a single arm (blob) of the 'A', separated by 55 Å from stem I. Due to the low resolution of SAXS model, the stem-loop II arm does not provide any structural features (such as 3-way junction) that can be compared to our structure. However, our structure and binding assays suggest that the RRE-Rev binding model, where the Rev dimer binds across the 'A' (i.e., across stem II and I), is unlikely. We added our interpretation in the Discussion.

"Previously, an RRE-Rev binding model was proposed based on small-angle X-ray scattering (SAXS) data of RRE (18). Because the spacing (~55 Å) between the stem II and I in the 'A'-shaped RRE model is similar to the spacing between the two RNA-binding domains in Rev dimers, an initial Rev dimer was proposed to bridge the stem II and I Rev-binding sites (IIB and IA sites) (18). Our Rev-binding model suggests that SLII alone binds the initial Rev dimer, and is thus different from the SAXS-based model."

Minor comments

1. Text requires careful proofreading for some writing issues such as missing words (e.g., Abstract, 3rd line up, "Based on these results and...."; Intro, 4 lines down "...the early stage of HIV infection,..."), failing to define some terms (nt, WT, etc) and a few grammatical errors.

We have corrected the errors.

2. Page 4, what is meant by "primes" in the context of Rev recognition? This is a vague term and not standard in the field.

We have changed 'primes' to 'is recognized by'.

3. Page 4 also has an example of a paragraph that is too long and could be split following ref 38, starting a new paragraph with "To better understand...."

We made a new paragraph.

4. Throughout the manuscript, the authors use "in vivo" when in fact they mean "in cells" as these were not animal studies.

We have corrected 'in vivo' to 'in cells'.

5. The authors should avoid "booster" words such as "drastically"

We have removed the word 'drastically'.

6. The apostrophes should be changed to prime symbols (e.g. page 8 the C1'-C1' distance).

We have corrected the apostrophes to prime symbols.

7. Methods: what is the source of T7 polymerase used? (should also be T7 RNA polymerase)

We used the plasmid obtained from Dreyfus lab and added the reference (68).

Sincerely,

Kay Choi, Ph.D.

Professor
Dept of Molecular and Cellular Biochemistry
Indiana University
Bloomington, IN47405

REVIEWERS' COMMENTS

Reviewer #1 (Remarks to the Author):

I am satisfied by the way the authors addressed my comments.
I believe this is an excellent paper that will make a huge impact on the HIV community.

Reviewer #2 (Remarks to the Author):

The authors have provided a very useful diagram in the supporting information of the crystal contacts in Figure S1. It would be helpful if the authors added a list of the crystal contacts. It appears to have some significant crystal contacts in the three-way junction in the open conformation. Could the authors model Rev binding without further conformational changes at these sites? Is it possible that these contacts provide the stabilization for the conformational changes of the syn G nucleotides? Either way, the structure and the results are useful from a biophysical point of view in understanding the forces stabilizing local and tertiary structure motifs in RNA.

Reviewer #3 (Remarks to the Author):

The authors have addressed my concerns.